# Network pharmacology and molecular-docking-based strategy to explore the potential mechanism of salidroside-inhibited oxidative stress in retinal ganglion cell

**Peng Zhang, Hongxin Zhao, Xiangping Xia, Hua Xiao, Chong Han, Zhibo You, Junjie Wang, Fang Cao** [ID] *

Department of Neurosurgery, Affiliated Hospital of Zunyi Medical University, Zunyi City, Guizhou Province, P. R. China

* caof@zmu.edu.cn

**Data Availability Statement:** All relevant data are within the manuscript and its Supporting Information files.

## Abstract

### Background

Salidroside (SAL), the main component of *Rhodiola rosea* extract, is a flavonoid with biological activities, such as antioxidative stress, anti-inflammatory, and hypolipidemic. In this study, the potential therapeutic targets and mechanisms of SAL against oxidative stress in retinal ganglion cells (RGCs) were investigated on the basis of in-vitro experiments, network pharmacology, and molecular docking techniques.

### Methods

RGC oxidative stress models were constructed, and cell activity, reactive oxygen species (ROS), and apoptosis levels were examined for differences. The genes corresponding to rhodopsin, RGCs, and oxidative stress were screened from GeneCards, TCMSP database, and an analysis platform. The intersection of the three was taken, and a Venn diagram was drawn. Protein interactions, GO functional enrichment, and KEGG pathway enrichment data were analyzed by STRING database, Cytohubba plugin, and Metascape database. The key factors in the screening pathway were validated using qRT-PCR. Finally, molecular docking prediction was performed using MOE 2019 software, molecular dynamic simulations was performed using Gromacs 2018 software.

### Results

In the RGC oxidative stress model in vitro, the cell activity was enhanced, ROS was reduced, and apoptosis was decreased after SAL treatment. A total of 16 potential targets of oxidative stress in SAL RGCs were obtained, and the top 10 core targets were screened by network topology analysis. GO analysis showed that SAL retinal oxidative stress treatment mainly involved cellular response to stress, transcriptional regulatory complexes, and DNA-binding transcription factor binding. KEGG analysis showed that most genes were mainly enriched in multiple cancer pathways and signaling pathways in diabetic complications,

**Funding:** P.Z. has received funding from Guizhou Provincial Department of Science and Technology, through the Science and Technology Program of Guizhou Province (ZK[2021]YB473]). The funders had no role in study design, data collection and analysis, decision to publish, or preparation of the manuscript.

**Competing interests:** The authors have declared that no competing interests exist.

nonalcoholic fatty liver, and lipid and atherosclerosis. Validation by PCR, molecular docking and molecular dynamic simulations revealed that SAL may attenuate oxidative stress and reduce apoptosis in RGCs by regulating SIRT1, NRF2, and NOS3.

## Conclusion

This study initially revealed the antioxidant therapeutic effects and molecular mechanisms of SAL on RGCs, providing a theoretical basis for subsequent studies.

## 1. Introduction

Retinal ganglion cells (RGCs) play an important role in visual communication and visual signal processing and control. As neuronal cells, RGCs are relatively large in size and length, with high energy requirements and a greater energy supply [1]. Under normal circumstances, the reactive oxygen species (ROS) generated during ATP production can be cleared by mitochondrial proteins and other scavengers to maintain oxygen balance [2]. However, when the mitochondrial function is impaired, the dynamic balance is disrupted, resulting in excessive ROS and then oxidative stress, damaging cell function, and possibly leading to cell death [3, 4]. Multiple studies have found that oxidative stress is one of the main mechanisms leading to various ophthalmic diseases, such as retinopathy. The mitochondrial dysfunction in glaucoma leads to impaired ATP synthesis in RGC, a significant increase in ROS, and significant death of RGC cells [5, 6]. In addition, the impairment of endoplasmic reticulum function leads to ROS accumulation and then oxidative stress [7, 8]. OPA1 mutation in hereditary optic atrophy leads to mitochondrial dysfunction and ROS accumulation [9]. In ischemic optic neuropathy, two peaks of ROS generation occur after ischemia–reperfusion injury. ROS accumulation can eventually lead to the destruction of intracellular oxygen balance and induce cell apoptosis [10, 11].

*Rhodiola rosea*, a perennial herb grown at high altitude, has been used for thousands of years in East Asia to treat cardiopulmonary disease and encephalopathy. Salidroside (SAL), a major component of *R. rosea* extract, is a flavonoid with biological activities, such as antioxidative stress, anti-inflammation, and lipid lowering; it has been found to have beneficial effects in psoriasis [12], atherosclerosis [13], cerebral spinal ischemia–reperfusion injury [14, 15], osteoporosis [16], and fatty liver [17]. It is also promising and valuable in the treatment of diseases, such as nonalcoholic hepatitis [18]. However, whether SAL has protective effect against oxidative stress in retinopathy is unknown, and studies on the related targets and molecular mechanisms are relatively lacking, which necessitates the application of big data to explore the existing targets and pathways of SAL related to oxidative stress in RGCs.

Therefore, in this study, network pharmacology, molecular docking, and PCR were used to explore the possible molecular mechanisms of action by examining the therapeutic effects of SAL on RGC oxidative stress and to provide a reference basis for subsequent studies. The protocol design is shown in Fig 1.

## 2. Materials and methods

### 2.1 Materials and reagents

RGC cells were purchased from Jennio Biotech Co., Ltd (Guangzhou, China). SAL medication was purchased from Merck KGaA (Darmstadt, Germany). DMEM/F12 medium and 0.25%

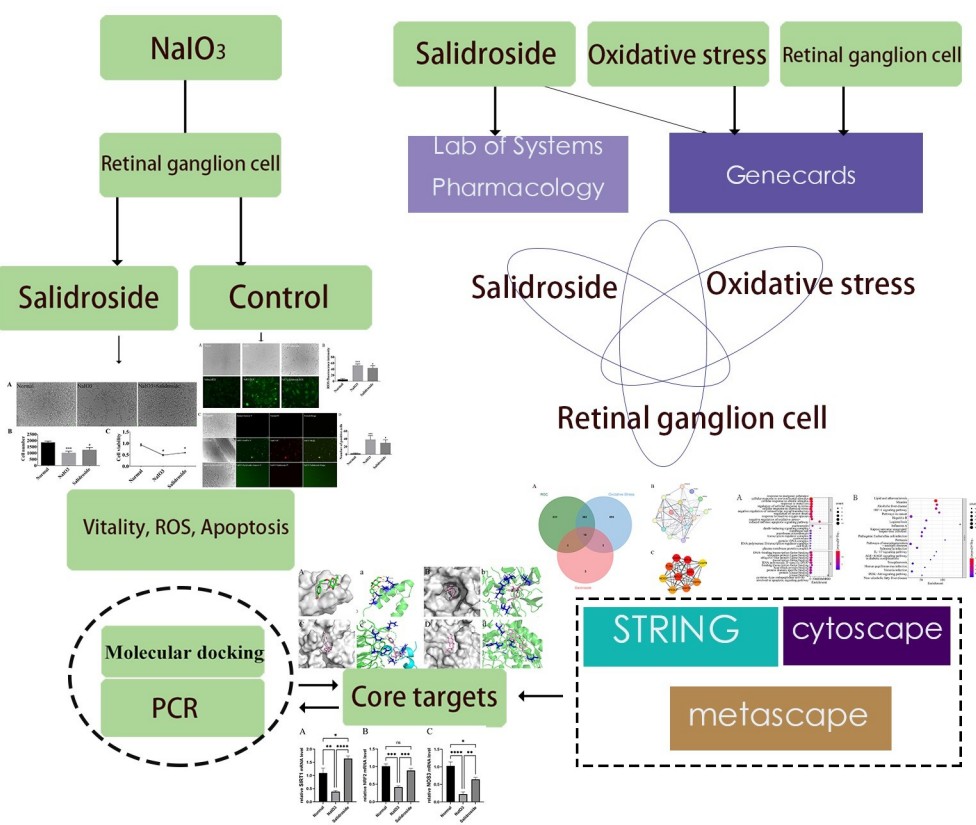

**Fig 1. Idea and process of this research.**

trypsin were purchased from Gibco (New York, USA). DCFH-DA assay kit was purchased from Solarbio Technology Co., Ltd (Beijing, China). Annexin V-PI apoptosis and necrosis assay kit were purchased from Beyotime Biotechnology Co., Ltd (Shanghai, China). CCK-8 cell viability assay kit was purchased from Bioss Biotechnology Co., Ltd (Beijing, China).

## 2.2 Experimental grouping

RGC cells was incubated in a humidified incubator at 37˚C with 5% $CO_2$. The experimental groups were as follows: normal control group (cultured normally without any intervention); $NaIO_3$ group (1200 UG/ml sodium iodate was added to induce for 24 h, and then the solution was replaced with DMEM/F12 containing 10% fetal bovine serum for further 48 h); and SAL group (the cells were first added to a concentration of 1200 UG/ml sodium iodate for 24-h induction, and then they were incubated for an additional 48 h with a mixture of SAL and DMEM/F12 containing 10% fetal bovine serum).

## 2.3 Sodium iodate-induced cell injury model and SAL pharmacological intervention

The RGC cell suspension was seeded into 96-well plates at $1 \times 10^4$ cells/well, with three wells for each group, and then cultured in DMEM/F12 medium containing 10% fetal bovine serum under 5% $CO_2$ at 37˚C. When the cell density reached 80%, the cells were group in accordance with 2.2. Among them, the normal control group was cultured normally without any intervention. In the $NaIO_3$ group, 1200 UG/ml sodium iodate was added to induce for 24 h, and then

the culture was continued with DMEM/F12 containing 10% fetal bovine serum for 48 h. The cells in the SAL group were first added to a concentration of 1200 UG/ml sodium iodate for 24-h induction and then incubated with a mixture of SAL and DMEM/F12 containing 10% fetal bovine serum for another 48 h. The cells in each group were randomly photographed from three fields and counted. The cell number changes, morphological changes (cell number and size), and other indicators between the experimental and control groups were compared.

## 2.4 ROS detection

The DCFH-DA method was used for ROS detection. The RGC cells were seeded into 96-well plates at $1 \times 10^4$ cells/well, with three wells for each group, and incubated at 37˚C in an incubator with 5% $CO_2$. They were treated as described in 2.2 to achieve 80% confluence. Afterwards, the medium was removed, and the cells were washed two times with PBS. Then, 10 μmol/L of the DCFH-DA working solution was added at 200 μl/well, and the cells were incubated at 37˚C for 30 min. Next, the medium was removed, and the cells were washed two times with PBS. Subsequently, 100 UL of the culture solution was added to each well, which was then observed under a fluorescence microscope. Three perimetry fluorometer averages were randomly photographed from each group, and then the content of cellular ROS between the groups was compared.

## 2.5 Annexin V-PI apoptosis assay

The RGC cells were seeded into 96-well plates at $1 \times 10^4$ cells/well, with three wells for each group, and incubated in an incubator at 37˚C and 5% $CO_2$ until 80% cell confluence was reached. Then, the cells were treated in accordance with the grouping described in 2.2. The medium was removed by aspiration. Next, the cells were washed one time with PBS, followed by the addition of 195 μl Annexin V-PI binding solution, 5 μl Annexin V-FITC, and 10 μl PI. The cells were observed under a fluorescence inverted microscope after being incubated for 10–20 min in the dark at room temperature (20˚C–25˚C).

## 2.6 CCK-8 cell viability assay

The RGC cells were seeded into 96-well plates at $1 \times 10^4$ cells/well, with three wells for each group, and incubated in an incubator at 37˚C and 5% $CO_2$ until 80% cell confluence was reached. After they were treated in accordance with 2.2, the CCK-8 cell viability assay reagent was added for 2 h. The absorbance values (OD values) at 450 nm were measured by a microplate reader, and three wells were repeated for each cell type. The OD values were used for statistical analysis.

## 2.7 Screening of potential targets and selection of intersection

The keywords "retinal ganglion cell" and "oxidative stress" were searched in GeneCards database (https://www.genecards.org/). The targets of SAL were retrieved using the traditional Chinese medicine systems pharmacology and analysis platform (TCMSP) and GeneCards. The drawing tool of a website (http://www.bioinformatics.com.cn) was used to draw Venn diagrams for the coexisting intersection targets of RGC, oxidative stress, and SAL.

## 2.8 Construction of Protein–Protein Interaction (PPI) network and screening of core targets

Intersection genes were imported into STRING database (https://cn.string-db.org/), defining the species as *Homo sapiens*. PPI networks were retrieved and constructed. The nodes in the

network graph represent the targets, and the interactions between nodes are represented by edges. The exported PPI TSV-formatted files were subsequently imported into Cytoscape version 3.7.2, and the network interaction relationships were topologically analyzed using the software's cytohubba plugin. The top 10 key targets were selected in accordance with the Maximal Clique Centrality (MCC) algorithm.

## 2.9 GO functional enrichment and KEGG pathway enrichment analysis

By using the Metascape database (https://metascape.org/), Gene Ontology (GO) functional enrichment analysis and Kyoto Encyclopedia of Genes and Genomes (KEGG) pathway enrichment analysis were performed on the total differentially expressed protein dataset, upregulated differentially expressed protein dataset, and downregulated differentially expressed protein dataset. The species *Homo sapiens* was selected with a minimum number of protein enrichments of 2 and a p value set at 0.05. Due to the large number of enrichment results, the top 10 results (less than 10 entries for all) of the most significant GO-biological processes (BP), GO-cellular components (CC), and GO-molecular functions (MF) and the top 20 results (less than 20 entries for all) of most significant KEGG pathway by p values were selected for visualization. The visualization was based on the applied microson signaling platform (http://www.DMoinformatics.com.cn, a free online platform for data analysis and visualization).

## 2.10 PCR validation of core molecules

Synthesize cDNA according to the instructions of the kit. The OD260/OD280 ratios of the extracted RNA samples were between 1.8 and 2.0. Real-time quantitative PCR was subsequently performed, and the primer sequences are shown in S1 Fig. The reaction conditions were as follows: initial denaturation at 95˚C for 5 min; annealing at 55˚C for 30 seconds; and extension at 72˚C for 1 min, followed by 30 cycles of denaturation at 95˚C for 10 seconds, annealing and extension at 72˚C for 10 min, and storage at 4˚C. The relative expression of the core target genes was assessed using the $2^{-\Delta\Delta CT}$ method. GAPDH was used as the internal reference to amplify the genes of interest.

## 2.11 Core molecular docking validation

The protein 3D profiles of the core target genes screened were downloaded from the PDB database (https://www.rcsb.org/) to perform molecular docking between the core target genes screened by network pharmacology combined with PCR experiments on SAL. In addition, the 3D structure of the SAL molecule was downloaded from the PubChem database (https://pubchem.ncbi.nlm.), and the open software Babel was used for format conversion preprocessing. The molecular structures of small molecules and receptor proteins were pretreated with and without water and ligand removal by using PyMOL2.5, and docking was performed using MOE2019 before visualization using PyMOL2.5.

## 2.12 MD simulations

Gromacs2018 was selected as the dynamics simulation software. A water box of size 10*10*10 nm$^3$ was established, and an ion automatic equilibrium system was added. The steepest descent method was used to minimize the energy with the maximum number of steps (50,000 steps). The canonical system (NVT) and the isothermal and isobaric system (NPT) were used to balance the system, and then a 100ns MD simulation was performed at room temperature and pressure. The V-rescale temperature coupling method was used to control the simulation temperature to 300 K, and the Berendsen method was used to control the pressure to 1 bar.

The gmx_MMPBSA plugin in the gromacs program was used to calculate the binding free energy.

### 2.13 Statistical analysis

All experimental groupings were repeated with 3 replicate wells, and the metrology data were expressed as X ± s, and SPSS version 26.0 statistical software was applied for data analysis. Comparisons between two groups were analyzed by independent samples t-test, and one-way ANOVA was applied for comparisons among multiple groups. $P < 0.05$ was considered statistically significant.

## 3. Results

### 3.1 Effect of SAL on RGC oxidative stress

Compared with the normal group, the cells in the NaIO3 group shrunk and rounded, decreased in number, and the viability was reduced to less than 60% of the original, and the difference was statistically significant ($P < 0.05$), which indicated that the experimental model modeling was successful in this study. However, compared with the NaIO3 group, the cell number was increased and the viability was elevated in SAL group, the difference was statistically significant ($P < 0.05$), that is, SAL could alleviate the damage caused by sodium iodate on RGC cells, and play a role in treating the oxidative damage of RGC cells (Fig 2A–2C).

### 3.2 RGC ROS and apoptosis detection

The RGC cells subjected to pharmacological induction with sodium iodate showed a significant decrease in cell viability, a significant increase in ROS levels, and enhanced expression (Fig 3A), with statistically significant differences ($P < 0.05$, Fig 3B). Compared with the normal group, the $NaIO_3$ group exhibited enhancement in green fluorescence, and the difference was statistically significant ($P < 0.05$). However, the expression of ROS in RGC body was lower in the SAL group than in the $NaIO_3$ group, and the difference was statistically significant ($P < 0.05$). The findings demonstrated that SAL was able to alleviate the damage caused by sodium iodate on RGC cells, thereby playing a role in treating RGC cell oxidative damage. After the RGC cells were induced by sodium iodate, the cell survival rate significantly decreased and the cell apoptotic rate increased (Fig 3C) according to the results of Annexin V staining, with statistically significant difference ($P < 0.05$). However, the apoptotic rate of RGC cells in the SAL group significantly decreased compared with that in the $NaIO_3$ group, and the difference was statistically significant ($P < 0.05$, Fig 3D). These findings demonstrated that SAL was able to alleviate the damage caused by sodium iodate on RGC cells, thus playing a role in treating RGC cell oxidative damage.

### 3.3 PPI and screening of core targets

Among the data downloaded from GeneCards, a total of 1229 targets were selected for high RGC correlation, 1093 targets were screened for high-oxidative-stress correlation, and 21 targets were screened for high-SAL correlation. The 16 common targets were taken using the Venn diagram tool in the microson letter platform for the subsequent data analysis (Fig 4A). The final intersection targets were subjected to PPI analysis and construction using the STRING website. The results showed that 76 interaction relationships existed among 16 targets (Fig 4B). Meanwhile, the top 10 core target proteins identified on the basis of the MCC algorithm by using the cytohubba plugin in Cytoscape version 3.7.2 were SIRT1, HIF1A, CASP3, MTOR, TLR4, GSK3B, HSPA5, NRF2, NOS3, and CASP8 (Fig 4C).

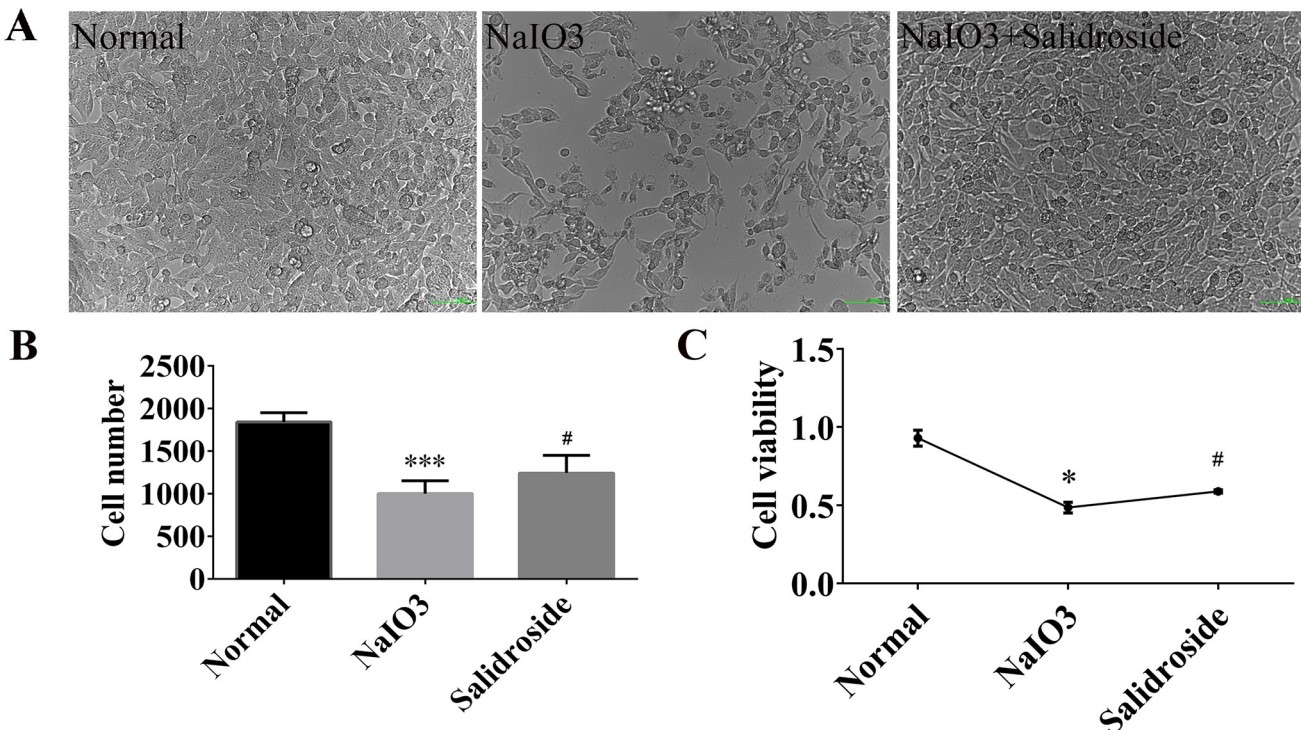

**Fig 2. Bright field view (20x) of cells in each group.** (A)Bright field perimetry plots of each group; (B)Comparison of cell count results among groups; (C) CCK-8 cell viability values of each group. The X-axis and Y-axis represent the group name and cell viability, respectively. *Statistically significant difference compared with the control group. #Statistically significant difference compared with the NaIO$_3$ group.

### 3.4 GO and KEGG enrichment analysis

The GO functional analysis results were screened in terms of p-value from small to large. The five most highly enriched BP terms were mainly involved biological functions, such as response to inorganic substances, cellular response to abiotic stimuli, cellular response to environmental stimuli, response to metal ions, and regulation of cellular response to stress; The CC terms included euchromatin, transcriptional regulatory complexes, RNA polymerase II transcriptional regulatory complexes, death inducing signaling complexes, membrane rafts; In addition, highly enriched MF terms were DNA binding transcription factor binding, ubiquitin protein ligase binding, ubiquitin-like protein ligase binding, transcription factor binding, RNA polymerase II specific DNA binding transcription factor binding, heat-shock protein binding, and protein domain specific binding (Fig 5A). The KEGG pathway analysis results showed that most of the genes were mainly enriched in pathways related to colorectal cancer and signaling pathways with roles in diabetic complications, gastric cancer, nonalcoholic fatty liver disease, lipids and atherosclerosis, hepatitis B, inflammatory bowel disease, and other pathways (Fig 5B).

### 3.5 RT-PCR validation of core targets

The mRNA levels of SIRT1, Nrf2, and NOS3 in RGC cells subjected to oxidative stress significantly decreased compared with those in normal controls. However, SAL treatment can effectively increase these expression levels. SAL can significantly enhance the expression of most of the abovementioned related factors compared with the model group (P < 0.01). The

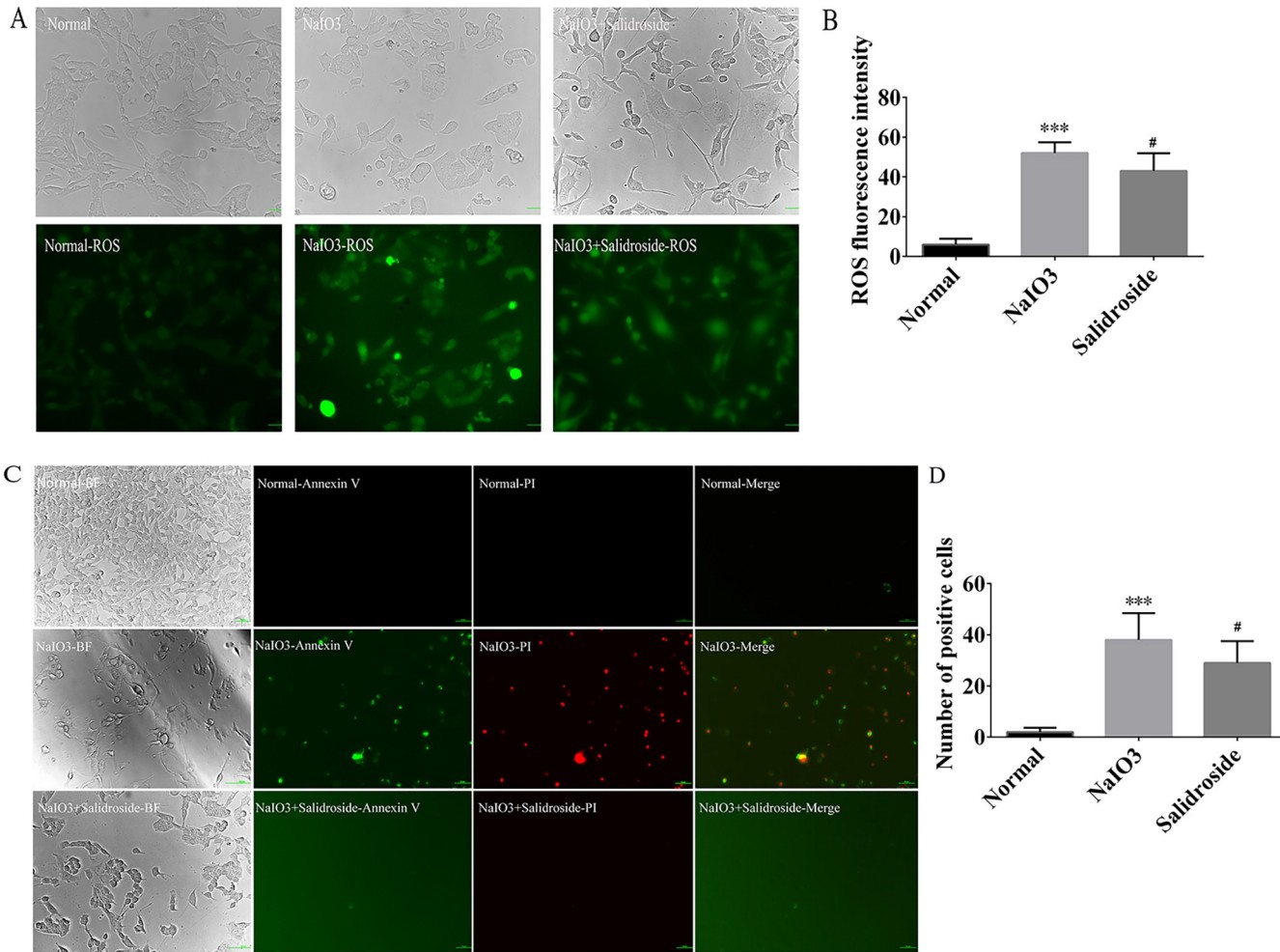

**Fig 3. Cellular reactive oxygen species (ROS, 20x) and apoptosis assays.** (A) Fluorescence expression diagram of ROS in each group; (B) Plots of fluorescence intensity quantification results of ROS in each group; (C) Graphs of Annexin V-PI fluorescence expression in each group; (D) Quantitative histograms of Annexin V-PI fluorescence number in each group. The X-axis and Y-axis represent the group name and the number of fluorescent spots respectively. *Statistically significant difference compared with the control group. #Statistically significant difference compared with the NaIO$_3$ group.

expression of Nrf2 and NOS3 in the SAL group was significantly higher than that in the model group and closer to that in the normal control group. Meanwhile, the expression of SIRT1 was significantly higher than that in the normal control group (Fig 6).

## 3.6 Molecular docking study

Molecular docking was performed to evaluate the affinity of drug candidates for their targets. The binding poses and interactions of SAL with the core targets SIRT1, Nrf2, and NOS3 proteins were obtained. The results showed that SAL can bind to SIRT1 protein targets through 5visible hydrogen bonds and strong electrostatic interactions. SAL binds to the Keap1/Nrf2 complex through 10 visible hydrogen bonds and strong electrostatic interactions. It can bind to the NOS3 protein target through 3 visible hydrogen bonds and strong electrostatic interactions. Moreover, the hydrophobic pockets of each target were successfully occupied by SAL (Fig 7).

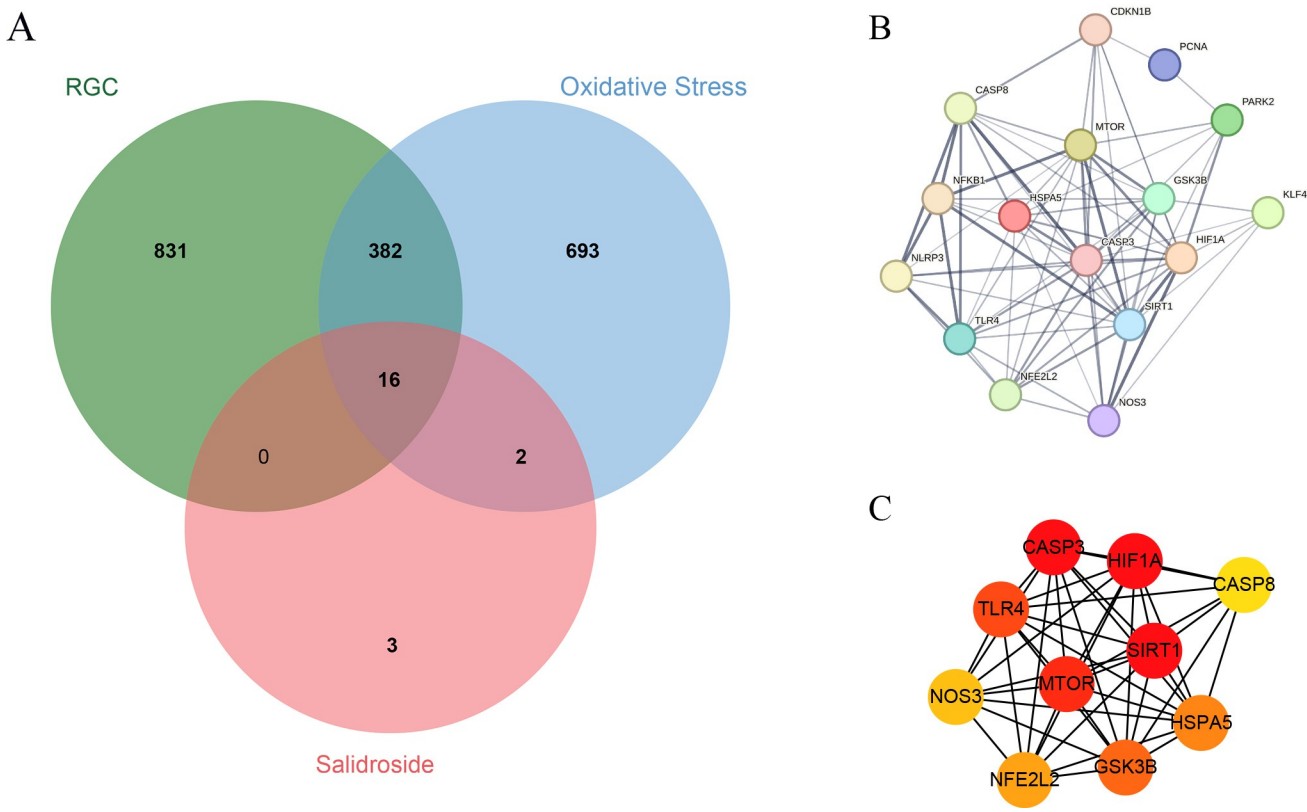

**Fig 4. PPI and screening of core targets.** (A) Venn diagram of coexisting intersection targets of RGC, oxidative stress, and SAL. RGCs are shown in green, oxidative stress is shown in blue, and SAL is shown in red; (B) PPI network diagram of common targets; (C) Key target network whose color is MCC value mapping.

### 3.7 MD simulations

In the case of a protein–ligand complex, the radius of gyration (Rg) is used to characterize the overall size and shape of the complex. The Rg values of all the protein-ligand complexes remained stable during 100 ns simulations. The Rg value of SIRT1 showed a slight fluctuation

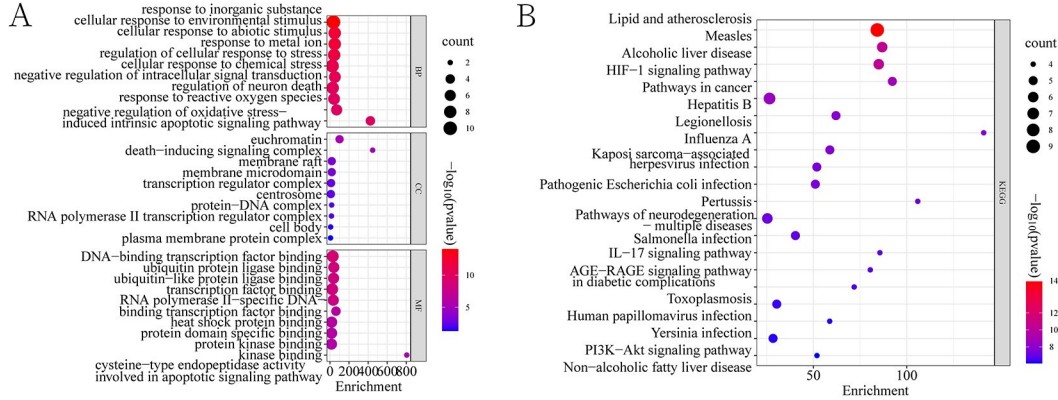

**Fig 5. Consensus target functions and pathway bubble plots.** (A) GO functional bubble plot of common targets; (B) Bubble plot of KEGG signaling pathways for common targets. The ordinate refers to GO/KEGG terms, the abscissa denotes fold enrichment, and the size of the dot indicates the number. The red color of bubble represents the p value.

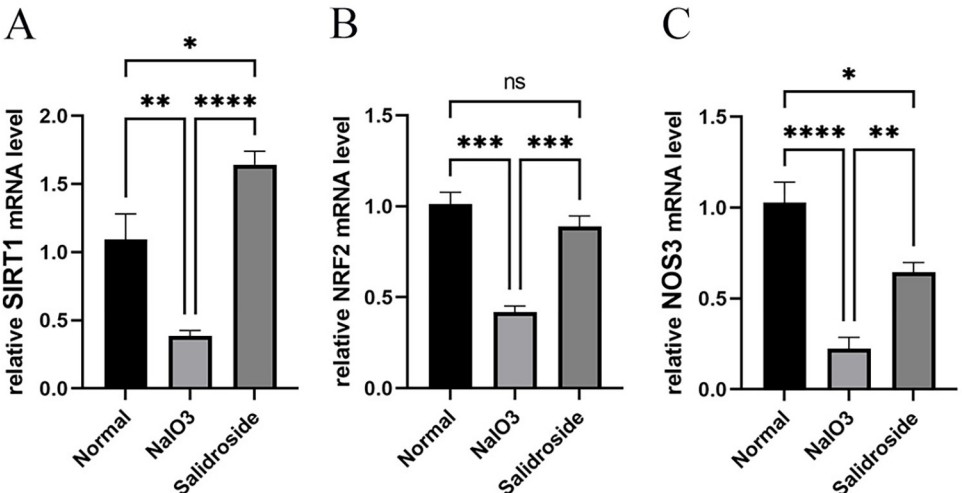

**Fig 6. Detection of mRNA expression of SIRT1, Nrf2, and NOS3 in each group.** Statistical differences were as follows: * P < 0.05, ** P < 0.01, *** P < 0.001, **** P < 0.0001.

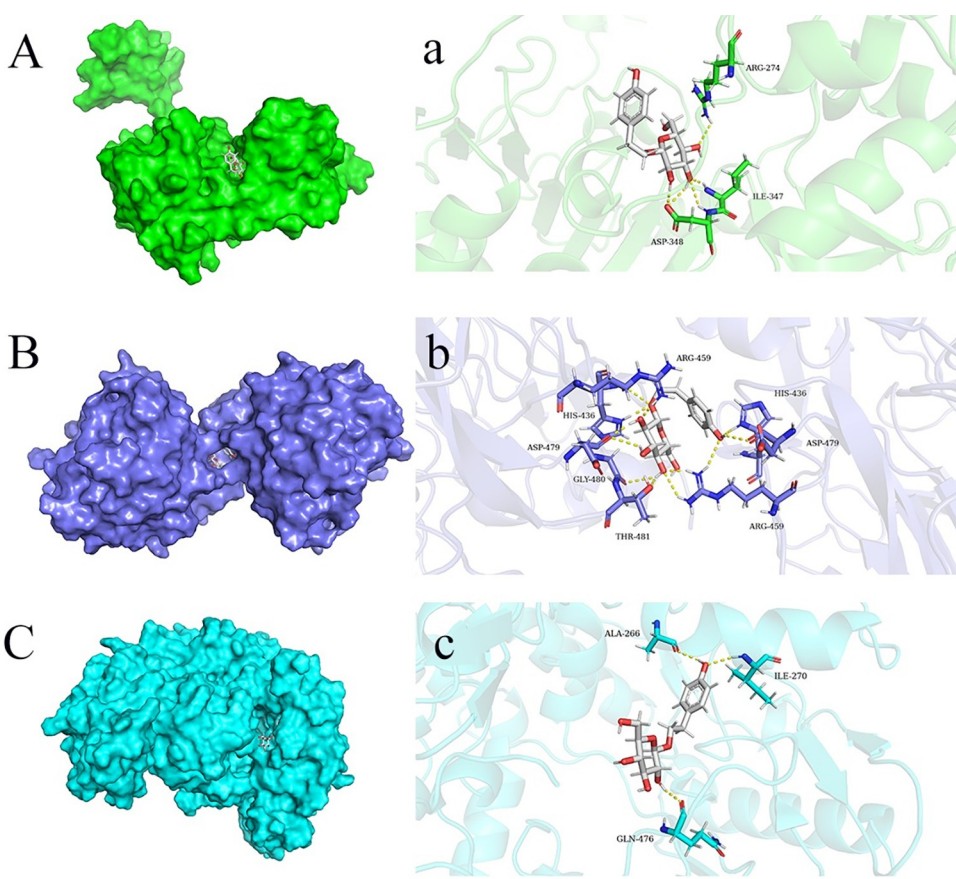

**Fig 7. Screening of drugs binding to their targets by molecular docking.** (A) Binding pocket of SAL to SIRT1, (a) interaction between SAL and target; (B) Binding pocket of SAL to Keap1/Nrf2 complex, (b) interaction of SAL with targets; (C) Binding pocket of SAL to NOS3, (c) interactions of SAL with targets.

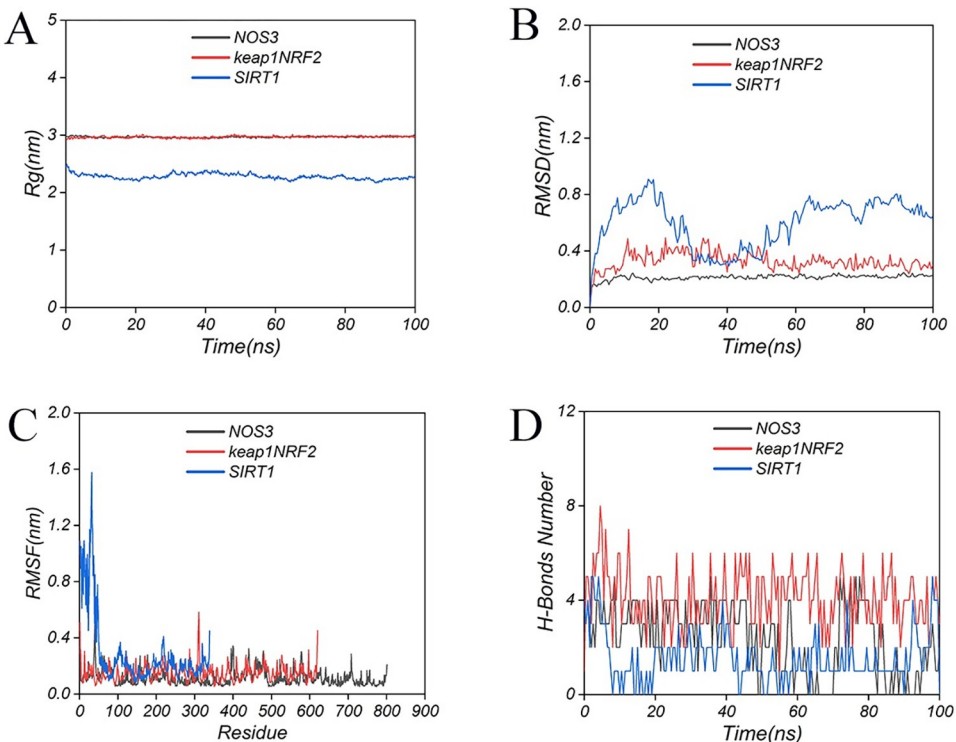

**Fig 8. MD simulations analysis of SAL in complex with proteins.** (A) Rg analysis of complexes; (B) RMSD analysis of complexes; (C) RMSF analysis of complexes; (D) The number of hydrogen bonds with time during 100 ns simulation.

compared to NOS3 and keap1NRF2, and the Rg values of the latter two remained stable throughout the simulation (Fig 8A). Root mean square deviation (RMSD) was used to observe the overall conformational changes of the protein relative to the initial structure during the simulation. Similar to the Rg trend, the RMSD value of the protein-ligand complex fluctuated slightly in the early stage of the simulation (SIRT1 fluctuated in the middle of the simulation), but as the simulation time increased, the RMSD value tended to stabilize, and finally the RMSD value stabilized at 0.227 nm (NOS3), 0.300 nm (keap1NRF2), and 0.647 nm (SIRT1) (Fig 8B). Root mean square function (RMSF) can be used to observe the structural fluctuations of local amino acid residue sites during the simulation process. In the NOS3 and keap1NRF2 systems, the RMSF values of all amino acid residues of the protein were < 0.8 nm, and most residues were stable. In the SIRT1 system, the RMSF of the first 50 residues of the protein was larger (Fig 8C). In addition, the time-dependent curves of the number of hydrogen bonds were analyzed. During the 100ns simulation, the average number of hydrogen bonds was 2.4 ±1.4 (NOS3), 4.1±1.2 (keap1NRF2), and 1.6±1.1 (SIRT1) (Fig 8D). The binding free energies of the three protein-ligand complex systems were analyzed. The final binding free energies of the ligand and protein were -12.63±6.27 kcal/mol (NOS3), -21.77±4.99 kcal/mol (keap1NRF2), and -12.00±6.03 kcal/mol (SIRT1), respectively (Fig 9).

## 4. Discussion

Oxidative stress damage is involved in important pathophysiological processes and closely related to the activation of inflammatory response; endothelial dysfunction; and autophagy or apoptosis in many ophthalmic diseases, such as glaucoma, diabetic eye disease, and optic nerve

### Binding free energy of three protein-ligand complex systems

| Mol ID | Chemical formula | Molecular weight | Docking protein | Binding energy/(kcal·mol−1) |
|---|---|---|---|---|
| MOL002929 | C14H20O7 | 300.3g/mol | SIRT1 | -12.00 ± 6.03 |
| | | | NRF2 | -20.77 ± 4.99 |
| | | | NOS3 | -12.63 ± 6.27 |

**Fig 9. Binding free energies of the three protein-ligand complex systems.**

degeneration. So, obtaining an antioxidant drug to attenuate oxidative stress damage is important. SAL, a flavonoid active ingredient extracted from traditional Chinese herbal medicine, has been found to have good antioxidant and anti-inflammatory effects in multiple disease models by previous pharmacological studies. Exploring whether SAL could be used to treat retinopathy and its antioxidant molecular mechanism is important. In this study, an in-vitro RGC oxidative stress model was established, and the results showed that the cell viability increased and the ROS and apoptosis decreased after SAL treatment. The core targets were screened for PCR and molecular docking to predict the mechanism of SAL against oxidative stress in RGCs by network pharmacology, and the RGC antioxidant activity and possible molecular mechanism of SAL were explored.

RGC oxidative stress was inducted by $NaIO_3$ in vitro. The number of RGC cells in the SAL + $NaIO_3$ group significantly increased, and the cell morphology was closer to normal than that in the $NaIO_3$ group. Through the detection of intracellular ROS, the $NaIO_3$ + SAL was found to effectively reduce the expression of ROS in RGC cells in vivo and attenuate oxidative stress injury. As detected by Annexin V-PI apoptosis, the apoptosis of RGC cells effectively reduced in the $NaIO_3$ + SAL group and played a role in attenuating oxidative damage, which was similar to the results of multiple other studies. Xiong Yanlei established hypoxic hepatitis in SD rats through a decompressing chamber and found that treatment with SAL can significantly inhibit inflammatory factor release and ROS and MDA production. In addition, SAL enhanced Nrf2-mediated activation of antioxidant pathways and inhibited JAK2/STAT3-mediated release of proinflammatory factors [19]. Wang Y used human umbilical endothelial cells to mimic a cerebral artery endothelial model, causing inflammation and injury by using hypoxic glucose deprivation and recovery. Treatment with SAL selectively increased complements C3 and C3a in endothelial cells without increasing the complement in astrocytes and microglia, resulting in the inhibition of inflammatory response [20]. Yao Fei established a mouse model of diabetic retinopathy, and long-term treatment with SAL alleviated elevation in blood glucose and lipid levels, reduced the level of oxidative stress in the retina, repaired diabetes-induced transcriptome abnormalities, and attenuated diabetic fundus microangiopathy [21]. Tang Yan established an oxidative stress model in PC12 cells, and after treatment with SAL, LDH decreased and SOD/GSH PX activities increased. The mechanism of increased mitochondrial energy synthesis, alleviated mitochondrial swelling and dissipation of mitochondrial membrane matrix, protected mitochondria, and inhibited apoptosis may be related to the elevation of HIF1A, ISCU1/2, and COX10 levels by SAL [22]. In conclusion, SAL has therapeutic effects in various diseases, and the mechanism may be associated with the inhibition of oxidative stress and inflammatory responses, etc.

Sixteen potential targets of oxidative stress in SAL reference RGCs were initially identified by network data mining to investigate the specific molecular mechanism in the RGC oxidative stress model. PPI analysis was performed to construct the interaction relationship, and the top 10 core targets were simultaneously screened out for GO and KEGG enrichment analysis. In

biological processes, the genes were mainly enriched in the cellular response to inorganic agents, environmental stimuli, chemical stimuli, ROS, and oxidative stress induction. In cell fractions, they were mainly enriched in euchromatin, transcriptional regulatory complexes, membrane rafts, and soma. In molecular functions, they were mainly enriched in transcription factor binding and protease binding. The KEGG enrichment analysis found that the genes were associated with multiple pathways in cancer, non-alcoholic fatty liver disease, lipids and atherosclerosis, and inflammatory bowel disease. On the basis of RT-PCR validation and molecular docking, SIRT1, Nrf2 and NOS3 were speculated to be the key targets of SAL to ameliorate oxidative stress in RGCs.

SIRT1 is an important subunit member of the Sir2 protein family, which has been implicated as a regulator in aging, inflammation, stress, and metabolic regulation. In this study, SIRT1 was decreased in RGCs subjected to oxidative stress, which was reversed by SAL and attenuated RGC oxidative stress injury. Li Huimin established a murine colitis model, and SAL treatment attenuated colonic tissue damage. The mechanism may be that SAL inhibited oxidative stress by upregulating SOD/GSH-Px/CAT; downregulating the expression of apoptotic proteins, such as Bax/Caspase-3; and regulating the SIRT1/FoxOs signal transduction pathway, so the expression of SIRT1, FoxO1, FOXO3a, and foxo4 in colon tissue increased [23]. You Baiyang prepared an in-vitro and -vivo model of insulin resistance and found that SAL reduced insulin resistance; the mechanism may be that SAL activated the AMPK/SIRT1 signaling pathway to regulate mitochondrial quality control and ROS production and reduced insulin resistance [24]. Zhao Dong used ox-LDL to induce endothelial cell injury and found that SAL increased SIRT1 expression, enhanced cell viability, and reduced cell apoptosis; the mechanism may depend on the activation of the AMPK/SIRT1 pathway [25]. The PCR and molecular docking prediction results in the present study were similar to those of the above studies. SIRT1 may be an effective target of SAL, and appropriate activation can effectively alleviate oxidative stress damage through the FOXO pathway, in addition to inhibiting Bax, p53, NF-κB, and PPAR γ expression; alleviating apoptosis and inflammation; and ultimately achieving the purpose of protecting cells from oxidative damage. The Keap1-Nrf2 signaling pathway is currently recognized as an important defensive transduction pathway in the body against oxidative and harmful stimuli from the internal and external environments. Keap1 and Nrf2 generally exist in the cytosol as inactive dimers and dissociate and activate during oxidative stress to initiate antioxidant stress pathways, such as Nrf2/HO-1. In this study, excessive RGC oxidative stress caused the intracellular oxygen balance to break, and cell dysfunction or even death resulted in a decrease in the overall Nfr2 expression. Meanwhile, SAL increased Nrf2 expression, without statistical difference with the normal group.

Yao Yuyuan established a mouse model of Alzheimer's disease and found that SAL could protect axonal morphology and improve cognitive function. The mechanism may be related to SAL directly binding to Nrf2 and blocking its Keap1 ubiquitin-linked interaction to activate Nrf2 [26]. Ji Rui found that dihydrotestosterone-induced oxidative stress in granulosa-like tumor cell lines was ameliorated after SAL treatment by reducing apoptosis, ROS accumulation, and oxidative damage and decreasing the depolarization of mitochondrial membrane potential. The underlying mechanism is that SAL upregulates AMPK/Nrf2 and downstream antioxidant enzymes [27]. Li Fuyuan established a rat model of cerebral ischemia–reperfusion and found that treatment with SAL reduced the infarct rate; the mechanism may be through the inhibition of oxidative stress by upregulating the Nrf2/Trx1 signaling pathway, increasing the activity of antioxidant enzymes, reducing the production of Caspase-3 and Bax/Bcl-2 apoptotic proteins, and inhibiting neuronal apoptosis [28]. Yang Sixia established a mouse model of Alzheimer's disease and HT22 neuronal cell iron death in vitro and found that SAL treatment improved cell survival, reduce lipid peroxidation and ROS levels, and improved

mitochondrial ultrastructure; the mechanism may be related to the Nrf2/HO-1 signaling pathway [29]. Xiao Lingling constructed a mouse model of neuroinflammation and cognitive impairment, and found that treatment with SAL reversed cognitive dysfunction in mice, reduced the expression of apoptosis and related apoptotic genes, and alleviated neuronal injury and inflammation; the molecular mechanism is related to the SIRT1/Nrf2 pathway and autophagy activation [30]. These studies were similar to the present study in that SAL could promote dissociation of Nrf2 from the Keap1 complex; increase Nrf2 expression; initiate downstream hox-1, NOS3, CNP, t-PA, and other factors; enhance cellular antioxidant, anti-inflammatory, anti-vasodilatory, and antiplatelet biological activities; and reduce ROS production and apoptotic generation. Endothelial nitric oxide (NO) synthase NOS3, which can catalyze the generation of NO, is involved in the pathophysiology of several diseases. This study found that the expression of NOS3 under oxidative stress decreased compared with that in the normal group, whereas the expression increased after SAL treatment, which was beneficial to the survival of cells. Xing Shasha established a mouse model of atherosclerosis through high-fat diet, and treatment with SAL effectively improved the skin function and reduced the atherosclerotic lesion area; these beneficial functions may be attributed to NO production and AMPK/PI3K/Akt/NOS3 pathway activation [31]. Zhang Wenjuan established an endothelial oxidative stress dysfunction model and found that administration of *Lycium barbarum* polysaccharide upregulated the activities of EGFR, ErbB2, and PI3K/Akt/NOS3; inhibited oxidative stress; reduced ROS production; and protected against endothelial cell injury [32]. On the contrary, Deng Ziteng established a cerebral ischemia model in vivo and in vitro by using rats and found that leonurine has a protective effect on cerebral ischemic injury; the effect depended on inhibiting the NO/NOS system to alleviate oxidative stress [33]. Similar to these studies, SAL treatment increased NOS3 expression, in which increased NO could directly relax blood vessels, and inhibited ROS-attenuated oxidative stress injury and inflammation by indirectly activating Nrf2 to reduce apoptosis.

Molecular docking and MD simulation showed that SAL exhibited good activity with affinity for the three targets. Among them, SAL showed the highest binding ability to the Keap1/Nrf2 dimer. Zhang Youbo established a murine colitis model and found that pilocarpine treatment significantly improved colonic inflammation and induced Nrf2 expression. Molecular docking also found that pilocarpine binds to Keap1, and the drug was speculated to directly inhibit the Keap1–nrf2 interaction and upregulate the Nrf2-mediated antioxidant response [34]. Yao Huan established an in-vitro and -vivo myocardial ischemia–reperfusion model and found that *Panax notoginseng* saponins improved cardiomyocyte injury, increased cell viability, and reduced ROS production. Further experiments found that *P. notoginseng* saponins can disrupt the interaction of the two by blocking the Nrf2 binding site in the Keap1 protein, increasing the dissociation of Nrf2, and thus initiating the related antioxidant pathways [35]. Although not the same small molecule, the above results were still similar to the docking results of the present study. SAL showed strong binding to the Keap1/Nrf2 complex, which can upregulate Nrf2 expression and inhibit oxidative stress by blocking Keap1/Nrf2 interaction. Therefore, good therapeutic outcomes may be achieved by targeting SIRT1, Nrf2, and NOS3 target genes, thus warranting further investigation.

## 5. Conclusion

Overall, this study first validated the antioxidant and antiapoptotic effects of SAL in an in-vitro RGC oxidative stress model. Then, the relevant targets of SAL treatment on oxidative stress in RGCs were collected from public databases, and their correlations and possible molecular mechanisms were analyzed by network pharmacology, followed by validation and prediction

by in-vitro PCR experiments and molecular docking software. The results showed that SAL effectively attenuated oxidative stress and reduced the intracellular ROS production and apoptotic rate in RGCs. SIRT1, Nrf2, and NOS3 may be the key molecular targets of this agent against oxidative stress. This study provides a reference for further investigation of the mechanism of action of SAL on oxidative stress in the treatment of retinopathy. However, the findings need to be further confirmed using in-vitro and -vivo experiments and clinical trials.

## Supporting information

**S1 Fig. PCR primer sequence.**
(TIF)

**S1 File. Original data.** All raw data required to replicate the results of study were listed in this file.
(ZIP)

## Author Contributions

**Data curation:** Peng Zhang, Hongxin Zhao, Xiangping Xia, Fang Cao.

**Formal analysis:** Peng Zhang, Xiangping Xia.

**Funding acquisition:** Peng Zhang.

**Project administration:** Peng Zhang, Xiangping Xia.

**Software:** Hongxin Zhao, Chong Han, Zhibo You, Junjie Wang, Fang Cao.

**Validation:** Hua Xiao, Chong Han, Zhibo You.

**Visualization:** Peng Zhang, Hongxin Zhao, Hua Xiao, Chong Han, Fang Cao.

**Writing – original draft:** Peng Zhang.

**Writing – review & editing:** Peng Zhang, Hongxin Zhao, Zhibo You, Junjie Wang, Fang Cao.

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
