## [Decision Letter · Decision Letter 0]

1 Apr 2024

PONE-D-24-07716Network pharmacology and molecular-docking-based strategy to explore the potential mechanism of salidroside-inhibited oxidative stress in retinal ganglion cellPLOS ONE

Dear Dr. Cao,

Thank you for submitting your manuscript to PLOS ONE. After careful consideration, we feel that it has merit but does not fully meet PLOS ONE’s publication criteria as it currently stands. Therefore, we invite you to submit a revised version of the manuscript that addresses the points raised during the review process.

We look forward to receiving your revised manuscript.

Kind regards,

Yash Gupta, Ph.D.

Academic Editor

PLOS ONE

Journal Requirements:

4. Please ensure that you include a title page within your main document. You should list all authors and all affiliations as per our author instructions and clearly indicate the corresponding author.

5. We note that Figure 1 in your submission contain copyrighted images. All PLOS content is published under the Creative Commons Attribution License (CC BY 4.0), which means that the manuscript, images, and Supporting Information files will be freely available online, and any third party is permitted to access, download, copy, distribute, and use these materials in any way, even commercially, with proper attribution. For more information, see our copyright guidelines: http://journals.plos.org/plosone/s/licenses-and-copyright.

Additional Editor Comments:

There is a validation step required with MD simulations > 100 nano seconds and WT-MetaD simulations for >50 nano seconds for the docking results

Reviewers' comments:

Reviewer's Responses to Questions

**Comments to the Author**

1. Is the manuscript technically sound, and do the data support the conclusions?

Reviewer #1: Partly

Reviewer #2: Partly

2. Has the statistical analysis been performed appropriately and rigorously? 

Reviewer #1: Yes

Reviewer #2: I Don't Know

3. Have the authors made all data underlying the findings in their manuscript fully available?

Reviewer #1: Yes

Reviewer #2: Yes

4. Is the manuscript presented in an intelligible fashion and written in standard English?

Reviewer #1: Yes

Reviewer #2: Yes

5. Review Comments to the Author

Reviewer #1: I believe that the manuscript is in general sound and does come to reasonable conclusions. However, the molecular docking portion of the paper seems to be incomplete. Docking alone, via Autodock, does not provide enough adequate support for the conclusions in this manuscript. Follow-up with molecular dynamic (MD) will provide more information to support the authors claims, via free energy calculations, e.g. MM/PBSA. I have listed some publications for the authors review.

Ramírez, D.; Caballero, J. Is It Reliable to Take the Molecular Docking Top Scoring Position as the Best Solution without Considering Available Structural Data? Molecules 2018, 23, 1038. [Google Scholar] [CrossRef]

Kumari, R.; Kumar, R.; Lynn, A. g_mmpbsa—A GROMACS Tool for High-Throughput MM-PBSA Calculations. J. Chem. Inf. Model. 2014, 54, 1951–1962. [Google Scholar] [CrossRef]

Campanera, J.M.; Pouplana, R. MMPBSA Decomposition of the Binding Energy throughout a Molecular Dynamics Simulation of Amyloid-Beta (Aß10−35) Aggregation. Molecules 2010, 15, 2730–2748. [Google Scholar] [CrossRef]

Wang, C.; Greene, D.; Xiao, L.; Qi, R.; Luo, R. Recent Developments and Applications of the MMPBSA Method. Front. Mol. Biosci. 2018, 4, 87. [Google Scholar] [CrossRef]

Once free energy calculation have been performed,I think this paper will be more sound for publication.

Reviewer #2: Dear authors,

There are some major and minor concerns regarding the manuscript as described below:

1. Please check the formatting in manuscript thoroughly and add the spaces wherever needed.

2. Fig. 2c and 3d; mention X and Y axis descriptions in detail.

3. The format of references is different from the manuscript body. Please check it.

4. Use subscript in the chemical formula.

5. In Line 224 and 228; check the spelling of SAL.

6. In Material and method section; authors have referred “1.2.2” but there is no such explanation to it.

7. Elaborate the abbreviations used in the manuscript.

8. Some grammatical errors in Line 290 and 304.

6. PLOS authors have the option to publish the peer review history of their article (what does this mean?). If published, this will include your full peer review and any attached files.

Reviewer #1: No

Reviewer #2: No

---

## [Author Response · Author response to Decision Letter 0]

18 May 2024

Dear Editors and Reviewers: 

Thank you for your letter and for the reviewers’ comments concerning our manuscript 

entitled “Network pharmacology and molecular-docking-based strategy to explore the potential mechanism of salidroside-inhibited oxidative stress in retinal ganglion cell” (PONE-D-24-07716). Those comments are all valuable and very helpful for revising and improving our paper, as well as the important guiding significance to our researches. We have studied comments carefully and have made correction which we hope meet with approval. The reviewer comments are laid out below in italicized font and specific concern shave been numbered. Our response is given in normal font and changes/additions to the manuscript are given in red text. 

Responds to the reviewer’s comments:

Author response: Thank you for your review. We have modified the citation style and other content as required.

Author response: Thank you for your review. We made sure to provide the research grant number received in the Funding Information section.

Author response: We apologize for any omissions in the uploaded content. We have uploaded the original data that can reproduce the research results as supporting information files. If there are any missing data, please let us know in time.

4.Please ensure that you include a title page within your main document. You should list all authors and all affiliations as per our author instructions and clearly indicate the corresponding author.

Author response: Thank you for your feedback. We have added the title page to the main document as requested.

5.We note that Figure 1 in your submission contain copyrighted images. All PLOS content is published under the Creative Commons Attribution License (CC BY 4.0), which means that the manuscript, images, and Supporting Information files will be freely available online, and any third party is permitted to access, download, copy, distribute, and use these materials in any way, even commercially, with proper attribution. For more information, see our copyright guidelines: http://journals.plos.org/plosone/s/licenses-and-copyright.

Author response: Sorry for the oversight here, we have modified Figure 1 without changing the original meaning.

Additional Editor Comments:

There is a validation step required with MD simulations > 100 nano seconds and WT-MetaD simulations for >50 nano seconds for the docking results.

Author response: Thank you for your review. To solve this problem, three new authors joined our team. MOE2019 was used for molecular docking experiments. After docking the protein and small molecule, the optimal complex conformation was selected according to the binding score, and Gromacs2018 software was used for 100ns dynamic simulation. This part is supplemented in the methods and results section.

Reviewer #1: 

1.I believe that the manuscript is in general sound and does come to reasonable conclusions. However, the molecular docking portion of the paper seems to be incomplete. Docking alone, via Autodock, does not provide enough adequate support for the conclusions in this manuscript. Follow-up with molecular dynamic (MD) will provide more information to support the authors claims, via free energy calculations, e.g. MM/PBSA. I have listed some publications for the authors review.

Author response: Thank you for your review. To solve this problem, three new authors joined our team. MOE2019 was used for molecular docking experiments. After docking the protein and small molecule, the optimal complex conformation was selected according to the binding score, and Gromacs2018 software was used for 100ns dynamic simulation. This part is supplemented in the methods and results section.

Reviewer #2:

1.Please check the formatting in manuscript thoroughly and add the spaces wherever needed.

Author response: Thanks for your review, we added spaces on line 21 and elsewhere where necessary.

2.Fig. 2c and 3d; mention X and Y axis descriptions in detail.

Author response: Thank you for your suggestion, we have added the description at lines 258 and 259.

3.The format of references is different from the manuscript body. Please check it.

Author response: We have revised the format of the references to be consistent with the main text.

4.Use subscript in the chemical formula.

Author response: We modified the chemical formula in line 107.

5.In Line 224 and 228; check the spelling of SAL.

Author response: Thank you for your review, we have changed the spelling of SAL.

6.In Material and method section; authors have referred “1.2.2” but there is no such explanation to it.

Author response: Sorry for the writing error here, we have changed 1.2.2 to 2.2 and marked it in red font.

7.Elaborate the abbreviations used in the manuscript.

Author response: We have added the full name of the abbreviation in lines 183, 187, 188, 195, 196 and marked them in red font.

8.Some grammatical errors in Line 290 and 304.

Author response: Thank you for your review. We have modified 3.4 GO and KEGG Enrichment Analysis and marked it in red font.

Special thanks to you for your good comments.

---

## [Editor Report · Decision Letter 1]

23 May 2024

PONE-D-24-07716R1Network pharmacology and molecular-docking-based strategy to explore the potential mechanism of salidroside-inhibited oxidative stress in retinal ganglion cellPLOS ONE

Dear Dr. Cao,

Thank you for submitting your manuscript to PLOS ONE. After careful consideration, we feel that it has merit but does not fully meet PLOS ONE’s publication criteria as it currently stands. Therefore, we invite you to submit a revised version of the manuscript that addresses the points raised during the review process.

We look forward to receiving your revised manuscript.

Kind regards,

Yash Gupta, Ph.D.

Academic Editor

PLOS ONE

**Additional Editor Comments:**

Authors have done commendable Job revising the manuscript. Most of the comments have been addressed. RMSD of MD simulations does not reveal the binding stress for neighboring amino acid side chain residues. It is important to calculate the strain on the components of the receptor pocket to reach a conclusion regarding validity of the proposed protein ligand interaction. Even if authors cannot do WT-MetaD simulations they should at least perform mmPBSA calculations with their trajectories. It's quite simple to perform and is an adjunct analysis on already done simulations (Ref. 10.1021/acs.jctc.1c00645 & 10.1016/j.cpc.2014.09.010).

Free energy calculations validate the MD simulations and are a necessary step and are not difficult to perform.

---

## [Author Response · Author response to Decision Letter 1]

25 May 2024

Response to Reviewers

Dear Editors: 

Thank you for your letter and for the comments concerning our manuscript entitled “Network pharmacology and molecular-docking-based strategy to explore the potential mechanism of salidroside-inhibited oxidative stress in retinal ganglion cell”(PONE-D-24-07716R1). We have studied comments carefully and have made correction which we hope meet with approval. The reviewer comments are laid out below in italicized font and specific concern shave been numbered. Our response is given in normal font and changes/additions to the manuscript are given in red text. We have made changes to our financial disclosures and included an updated statement in the cover letter. 

Additional Editor Comments:

 Authors have done commendable Job revising the manuscript. Most of the comments have been addressed. RMSD of MD simulations does not reveal the binding stress for neighboring amino acid side chain residues. It is important to calculate the strain on the components of the receptor pocket to reach a conclusion regarding validity of the proposed protein ligand interaction. Even if authors cannot do WT-MetaD simulations they should at least perform mmPBSA calculations with their trajectories. It's quite simple to perform and is an adjunct analysis on already done simulations (Ref. 10.1021/acs.jctc.1c00645 & 10.1016/j.cpc.2014.09.010).

Free energy calculations validate the MD simulations and are a necessary step and are not difficult to perform.

Author response： Dear Editor, thank you for your suggestions. We have added the description of MMPBSA calculations in the Methods (2.12) section and made changes in the Results (3.6-3.7). The changes are marked in red font.

Special thanks to you for your good comments.

---

## [Editor Report · Decision Letter 2]

29 May 2024

Network pharmacology and molecular-docking-based strategy to explore the potential mechanism of salidroside-inhibited oxidative stress in retinal ganglion cell

PONE-D-24-07716R2

Dear Dr. Cao,

We’re pleased to inform you that your manuscript has been judged scientifically suitable for publication and will be formally accepted for publication once it meets all outstanding technical requirements.

Kind regards,

Yash Gupta, Ph.D.

Academic Editor

PLOS ONE